# Affordable Thin Lens Using Single Polarized Disparate Filter Arrays for Beyond 5G toward 6G

**DOI:** 10.3390/s19183982

**Published:** 2019-09-14

**Authors:** Inseop Yoon, Seongwoog Oh, Jungsuek Oh

**Affiliations:** 1Department of Electronic Engineering, Inha University, Incheon 22212, Korea; 2Institute of New Media and Communications and School of Electrical and Computer Engineering, Seoul National University, Seoul 08826, Korea

**Keywords:** lens, metasurface, millimeter wave antenna, polarization, spatial filter, transmit array

## Abstract

This paper proposes a novel design approach for a thin lens with the aim of overcoming fineness limits in the commercial millimeter wave printed circuit board (PCB) manufacturing process. The PCB manufacturing process typically does not allow the fabrication of metallic patterns with a gap and width of less than 100 μm. This hampers expanding thin lens technology to 5G commercial applications, especially when such technology is considered for 60 GHz or higher frequency, which requires a finer gap and width of metallic traces. This paper proposes that problematic process conditions can be mitigated when a lens is designed by establishing single-polarized lumped element models where larger capacitance and inductance values can be obtained for the same patch and grid unit cells. While the proposed design technique is more advantageous at higher target frequencies, a 60 GHz application and a wireless backhaul system is selected because of a limited range of frequencies that can be measured by an available vector network analyzer. The required gap or width of metallic traces can be widened significantly by using the proposed single-polarized unit cells to acquire the same in-plane capacitance or inductance. This enables the lens operating at higher-frequency under the process limits in fabricable fine traces. Finally, the effectiveness of the simulated design procedure is demonstrated by fabricating a 60 GHz thin lens that can achieve a gain enhancement of 16 dB for a 4 × 4 patch antenna array with a gain of 16.5 dBi.

## 1. Introduction

A backhaul system is defined as a system that delivers collected voice data to other base station backhaul systems or the backbone network of a mobile carrier. Recently, with the unexpected increase in smartphone usage, the importance of a backhaul system has increased. Thus far, a backhaul network is mainly established through wire connections at optical frequencies or wireless connections at the microwave band. Recently, the millimeter wave band backhaul system is attracting a considerable amount of attention because of advantages such as low construction, low maintenance costs, and the capability of being reconfigured when cooperating with massive wireless networks. A backhaul system with a frequency band of 60 GHz has been studied in 3GPP [1,2]. A millimeter wave band backhaul system requires a high antenna gain [3,4,5] and several studies have been proposed to realize a high antenna gain for backhaul systems [6,7,8,9]. Among the various methods of increasing antenna gain, a planar configuration of a millimeter wave thin lens is gaining popularity because of its simple design, system integration, and its practical geometrical advantages. Furthermore, as the backhaul system uses point-to-point communication between fixed backhaul antennas, it is appropriate to use a millimeter wave thin lens in areas where only fine steering is required rather than wide beam coverage.

Such a millimeter wave thin lens consists of spatially filtering unit cells providing various levels of phase shift. A conventional thin lens is implemented by combining frequency selective surfaces (FSSs) realized by metallo-dielectric structures such as patch unit cells or grid unit cells. Because the FSS correlates with the electrical length of the frequency, the size of a single unit cell tends to decrease as the frequency increases. Notably, the mass production of 60 GHz lenses is problematic because of manufacturing process conditions. While commercial PCB etching is the most realistic for such thin lens, the minimum gap and width of the metallic traces are just 100 μm when etching metal layers on the substrate with a tolerance of approximately 10%. Therefore, a patch or grid pattern smaller than the minimum limit is difficult to be manufactured by commercially available millimeter wave processes.

This study presents a novel design method to realize larger in-plane reactance values than those obtained by existing unit cell structure. This is realized by partially eliminating the gap (or strip) among patches (or grid) along one axial direction. It is found that the tunable range of the in-plane reactance values can be improved significantly when the thin lens is designed only for single-polarized incident waves. Finally, the detailed design procedure is described for a wireless backhaul lens antenna operating at 60 GHz, employing the proposed unit cell topologies.

## 2. Design of Single Polarized Thin Lens

### 2.1. Comparison of In-Plane Reactiance for Single/Dual Polarization

Conventional millimeter wave thin lenses are designed by utilizing a moderate tunable range of the phase shift acquired by different types of frequency selective surface (FSS) unit cells having low-pass or band-pass responses. Patch and grid elements, which are the most basic types of FSS, are shown in Figure 1a and Figure 2a. When such basic elements having sub-wavelength size are populated repeatedly, the capacitance and inductance in their equivalent circuit are determined by the in-plane coupling structure between the patches and the thin metal line structure of the grid element, respectively. With *y*-polarized incident waves, their equivalent circuits are shown in Figure 1c and Figure 2c. The theoretical capacitance and inductance values of the equivalent circuit can be obtained by Equations (1) and (2), respectively [10,11,12,13].
(1)$$C=\varepsilon_{0}\varepsilon_{eff}\frac{2d}{\pi }\ln\left(\frac{1}{\sin\left(\frac{\pi g}{2d}\right)}\right)$$
(2)$$L=\mu_{0}\mu_{eff}\frac{d}{2\pi }\ln\left(\frac{1}{\sin\left(\frac{\pi w}{2d}\right)}\right)$$

As can be seen from Equations (1) and (2), smaller values of *g* and *w* must be selected to obtain higher values of capacitance and inductance in a unit cell for a fixed value of *d*. Having higher capacitance and inductance values under the given physical dimensions is advantageous in two aspects. First, for the needed capacitance and inductance values wider gap and width of the unit cells can be used leading to reduced fabrication costs. Second, the lower resonant frequency can be realized when designing the low-pass and band-pass unit cells designed in Section 2.2. However, at frequencies greater than 60 GHz, a limit of minimum values of *g* and *w* may be a critical bottleneck to realize such higher values of capacitance and inductance under the commercial process conditions of the minimum available gap and width of 100 *μ*m or less. This challenge can be addressed moderately by designing the unit cells only for single-polarized incident waves. This approach transforms dual-polarized patches and grids having square and net topologies into one-dimensionally stretched strips, as shown in Figure 1b and Figure 2b. Figure 3 shows a comparison between the simulated capacitance values of the dual and single-polarized patch unit cells shown in Figure 1a,b. If the patch has a *g_s_* of 0.1 mm, the capacitance of the dual-polarized patch unit cell is approximately 35 *f*F, while that of the single-polarized patch unit cell is 53 *f*F, which is 1.5 times greater than that of the dual-polarized patch unit cell. For a *g_s_* of 0.15 mm, the capacitance increases from 28 *f*F to 42 *f*F. Similarly, Figure 4 shows that the single-polarized grid unit cell has a higher value of inductance compared with that of the dual-polarized grid unit cell. This is because the equivalent circuit of the net topology of the dual-polarized grid includes a shunt capacitance and the desired shunt inductance. Thus, the inductance is partially cancelled by the shunt capacitance. These characteristics validate that the proposed design approach can achieve higher values of capacitance and inductance for a unit cell without an increase in *d*, *g_d_*, and *w_d_*.

### 2.2. Single-Polarized Low-Pass and Band-pass Unit Cells

This section describes the geometry and frequency response of the selected unit cells with three metal layers to design a convex lens based on the capacitive and inductive strips proposed in Section 2.1. Figure 5 shows a side view of the three-metal-layer structure with two substrates and a bondply in the unit cell. Rogers RO3003 having a thickness of 127 μm is used as a substrate that has a dielectric constant and a loss tangent of 3.0 and 0.001, respectively. Rogers RO2929 is used as the bondply to bond the two substrates. The thickness, permittivity, and loss tangent of the bondply are 38 μm, 2.9, and 0.003, respectively. Figure 6 shows the low-pass and band-pass unit cells where all the strip elements are connected to the strips in the neighboring unit cells without any gap. If the type of unit cells constituting a lens are homogeneous, such as low-pass or band-pass unit cells, the tunable range of the phase shift is limited to be approximately 180° [14]. The easiest way to increase the tunable range is to increase the number of layers of the substrate. However, this may lead to an increase in cost, profile, loss, and a deterioration in performance due to increased alignment errors during the process [15]. To solve these problems, a design method to improve the tunable range by using both low-pass and band-pass unit cells is used [16]. In this sense, this paper investigates the proposed single-polarized unit cells under the same conditions as those of the unit cells with three metal layers. A low-pass unit cell consists of single-polarized strip unit cells in all three layers. A band-pass unit cell with an inductive strip placed at layer 2 and two capacitive strips located at layers 1 and 3 is designed.

Table 1 summarizes the geometry, insertion loss (IL), and phase of the ten spatially filtering unit cells selected for lens design from the unit cells having the aforementioned capacitive and inductive strips. A total of five low-pass and five band-pass unit cells are used to acquire a wide tunable range of the phase shift. Figure 7 shows the simulated transmittance of both low-pass and band-pass unit cells with different sized patches and different width grids. It can be seen that all unit cells are designed to have an IL of less than 1 dB at the target frequency of 60 GHz to minimize energy loss. Figure 8 shows the plots of the phase shifts of the selected unit cells according to the selected unit cells. It is observed that a tunable range of 274° at the target frequency of 60 GHz is acquired by combining the phase shifts of the appropriately tuned five low-pass and five band-pass unit cells.

### 2.3. Feeder Design: 4 × 4 Patch Antenna Array

Figure 9 shows the geometry of an antenna used as a feed source to verify the gain enhancement of the proposed lens with a given process limit. The antenna is designed as a rectangular patch, and the antenna uses the direct feed method through the via hole, as shown in Figure 9b. A 4 × 4 array configuration is selected for beam steering capability, which is an essential functionality for a wireless backhaul system requiring a slight beam adjustment of a few degrees in deployment scenarios. The designed patch antenna array is fabricated using a 10-layer FR4 stacked substrate, which is widely used in commercial millimeter wave antenna-in-package solutions because of the rapid development of 5G technology with a low manufacturing cost [17,18,19,20]. The thickness of the antenna substrate is 0.762 mm, and the dielectric constant and loss tangent are 4.2 and 0.002, respectively. All the 10 layers are filled with copper except for the areas related to dominant fringing fields, as shown in Figure 9a. This is because dielectric losses along the substrate increase substantially as the target frequency increase up to millimeter wave frequencies such as 60 GHz. Physical dimensions in the antenna array are *L_x_* = 16, *L_y_* = 16, *p_x_* = 1.04, *p_y_* = 1.04, *u_x_* = 1.9, *u_y_* = 1.9, *s_x_* = 2.5, and *s_y_* = 2.5 in millimeters (mm). Figure 10 shows the 3D radiation patterns of the designed 4 × 4 patch antenna array. Each beam pattern is formed when input phase offsets of 0, 45, 90, and 135° are applied along the *y*-axis. Accordingly, the beam pattern exhibits beam steering characteristics in the *θ*-direction according to the phase offset.

### 2.4. Lens Implementation Usnig Macro Design

This section describes the procedure and results of designing the proposed lens by using the selected unit cells and the feed antenna, described in Section 2.2 and Section 2.3, respectively. The design procedure is based on the general procedure described in [14,15,16,21], but modified for the proposed single-polarized lens.

Step 1:The proposed lens is chosen to be located at a distance of 150 mm away from the 4 × 4 patch antenna array. The diameter of the selected aperture for the lens is chosen to be 150 mm. The value of *f*/*D* was chosen to be 1.0, where f is the distance between the feed antenna and the lens, and *D* is the diameter of the lens, to implement a compact and focused design. It should be noted that the target distance also determines the adjustable range of the phase shift.Step 2:The phase profile of the waves emitted from the 4 × 4 patch antenna array source is captured on the selected aperture at the distance of 150 mm. In this case, the phase profile is captured every 1.5 mm from the center of the lens considering the size of the unit cell, which is 1.5 mm.Step 3:The phase values needed for collimating radiated rays at all captured points are calculated and matched to the selected unit cells in Table 1. For the interval outside the range of tunable range from −280 to 0°, LP1 and BP5 are used as alternatives.Step 4:If the type of the unit cell at each point is the same as that at the next point, the area formed by these two points is grouped and classified as a zone.Step 5:It should be noted that variations in the ILs over the lens aperture might have the same effect as that of the non-uniform magnitude of feeds in a phased antenna array, leading to beam deformation related to beam width, side lobes, and so on [22,23]. Table 2 lists the numbered zones, the number of selected unit cells in each zone, the required phase shift at the center of each zone, and the selected unit cells in each zone. In Table 2, it is assumed that the change in the phase shift due to the different incident angles of the unit cells constituting the proposed lens is not significant.Step 6:Finally, a macro design of the proposed lens is completed, as shown in Figure 11 through the aforementioned steps. The first and third layers consist only of the capacitive strip unit cells proposed in Section 2.2. In the second layer, both of the capacitive and inductive strip unit elements are repeatedly arranged according to the zone classified in step 5.

Figure 12 shows that the designed lens and the 4 × 4 antenna array are separated by a selected distance *f*. To achieve rapid full-wave simulations for the lens, a large-scale structure, unit cells are replaced by simple effective mediums. In Figure 13, the structure of the unit cell shown in Figure 12 is replaced by the effective mediums. 3D radiation pattern plots shown in Figure 14 are obtained by simulating the structure shown in Figure 13 using Ansys HFSS. Each beam pattern is plotted when the input phase offsets are set to be 0, 30, 60, and 90°, before and after mounting the lens. A gain enhancement of approximately 16 dB is obtained using the proposed lens along the boresight. In the absence of the lens, each peak beam is observed at 0, −10, −20, and −30°. The angles in which the peak beams are observed are changed to be 0, −1, −2, and −3° on adding a lens.

Trade-offs between gain enhancement and beam steering that essentially occurs in convex lenses are well known [24,25]. However, as a wireless backhaul system uses a beam steering coverage as small as a few degrees after antenna installation and maintenance, the presented coverage is allowable, as shown in Figure 15.

## 3. Fabrication and Measurement

Figure 16 shows the top view of the fabricated lens and the alignment keys used for precise alignment among the substrates during fabrication. Figure 17 shows the measurement setup for the proposed lens mounted by a jig. The antenna array designed in Section 2.3 is fabricated by adding a feed line that includes power dividers and phase shifting lines for each antenna element. The feedline is soldered by a connector with the model number of 18S102-40ML5 in Rosenberger company, for measurement. The manufactured 4 × 4 patch antenna array and lens are used as transmitters and the standard gain horn antenna of WR-15 type operating at 60 GHz is used as a receiver. Finally, the antenna array is connected to the Anritsu MS4647B Vector Network Analyzer to measure radiation patterns of the proposed lens antenna. The manufactured antenna and lens are mounted using a specially designed zig for fine alignment of the distance and roll between them.

The gain enhancement value obtained by the fabricated lens is extracted by comparing the received powers before and after mounting the lens in front of the antenna array. Figure 18 shows the simulated and measured results obtained by using a 4 × 4 patch antenna array before mounting the lens. The peak beams are respectively measured at 0, −9, −21, and −29° with phase offsets of 0, 30, 60, and 90°, and the gain values are 16.57, 16.46, 16.32, and 16.16 dBi, respectively. Figure 19 shows the simulated and measured results after mounting the lens. Setting the same phase offsets as the case of the antenna only, the peak beams of the proposed lens antenna are measured at 0, −1, −2, and −3°, respectively, and the gain values are 32.88, 31.87, 31.31, and 29.66 dBi, respectively. As a result, the measurement results appear to be in good agreement with simulation results.

## 4. Conclusions

This paper proposes a novel design approach to obtain higher values of capacitance and inductance with a limited fabricable gap and width. The proposed approach is validated by designing a high-gain thin lens at a high frequency of 60 GHz. To increase the tunable range of a phase shift up to 274°, a new class of disparate filter arrays that employ both low-pass and band-pass unit cells are proposed for single-polarized unit cells. A 60 GHz single-polarized thin lens applicable to a wireless backhaul system is designed, simulated, and fabricated for demonstrating the proposed design approach. The gain of a 4 × 4 patch antenna array is improved by approximately 33 dBi by using the proposed lens. This suggests the feasibility of applying the proposed antenna configuration to various 5G platforms, such as wireless backhaul systems and repeaters.

## Figures and Tables

**Figure 1 sensors-19-03982-f001:**
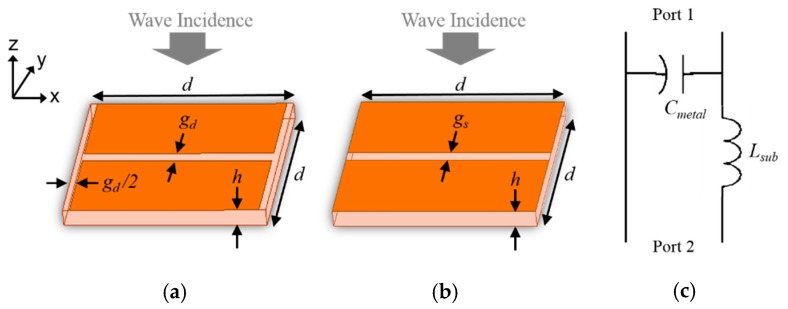
(**a**) Conventional dual-pol patch, (**b**) proposed single-pol patch, and (**c**) equivalent circuit of the patch.

**Figure 2 sensors-19-03982-f002:**
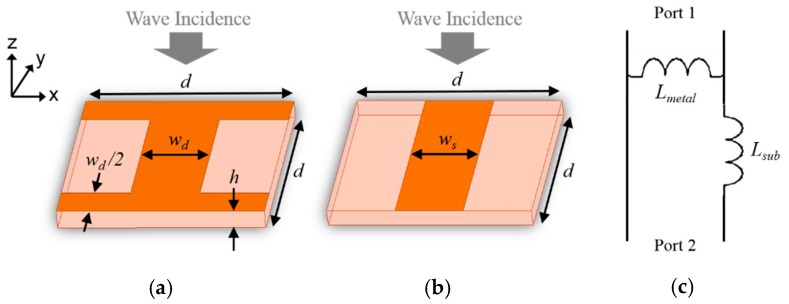
(**a**) Conventional dual-pol grid, (**b**) proposed single-pol grid, and (**c**) equivalent circuit of the grid.

**Figure 3 sensors-19-03982-f003:**
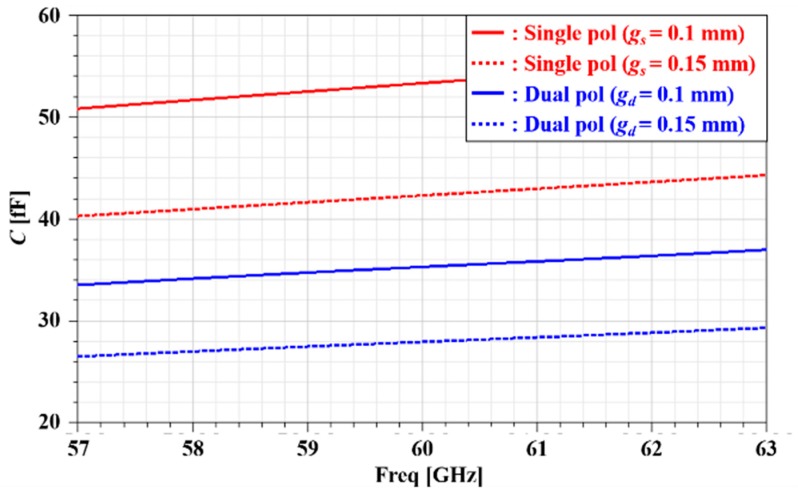
Simulated in-plane capacitance of dual-pol patch and single-pol patch.

**Figure 4 sensors-19-03982-f004:**
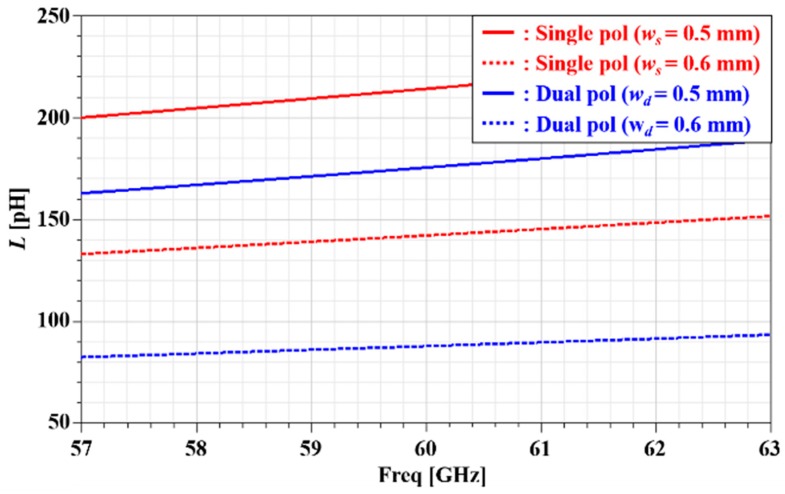
Simulated inductance of dual-pol grid and single-pol grid.

**Figure 5 sensors-19-03982-f005:**
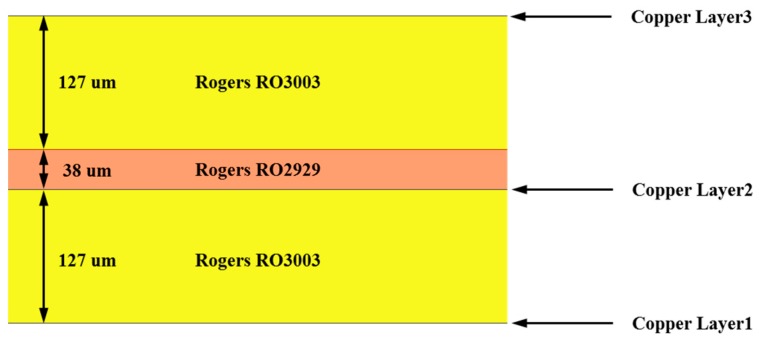
Side view of the three-metal-layer structure with two substrates and a bondply in the unit cell.

**Figure 6 sensors-19-03982-f006:**
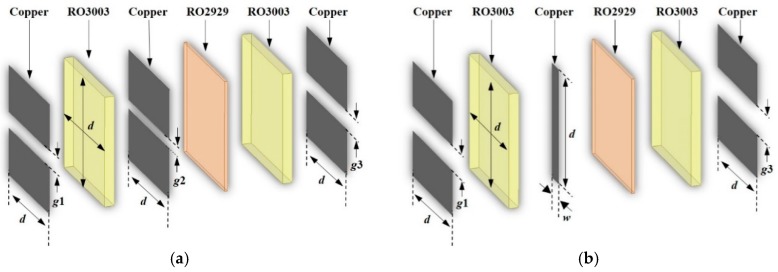
Exploded views and dimension parameters of single-pol, (**a**) low-pass, and (**b**) band-pass unit cells.

**Figure 7 sensors-19-03982-f007:**
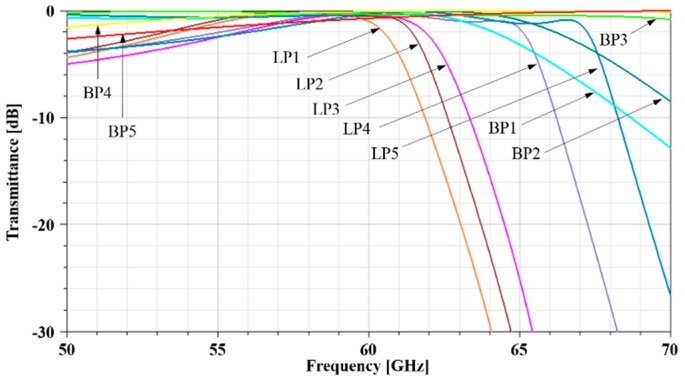
Simulated transmittance (s21) of both low-pass and band-pass unit cells with different sized patches and different width grid.

**Figure 8 sensors-19-03982-f008:**
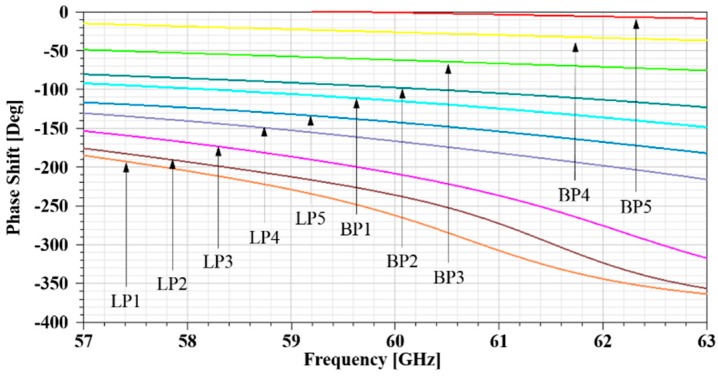
Simulated phase shift of both low-pass and band-pass unit cells with different patch sizes and grid width.

**Figure 9 sensors-19-03982-f009:**
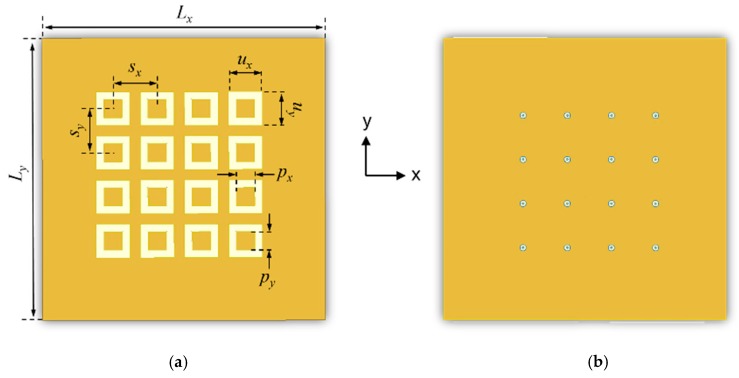
(**a**) Top and (**b**) bottom views of the 4 × 4 patch antenna array to use as a source for lens verification.

**Figure 10 sensors-19-03982-f010:**
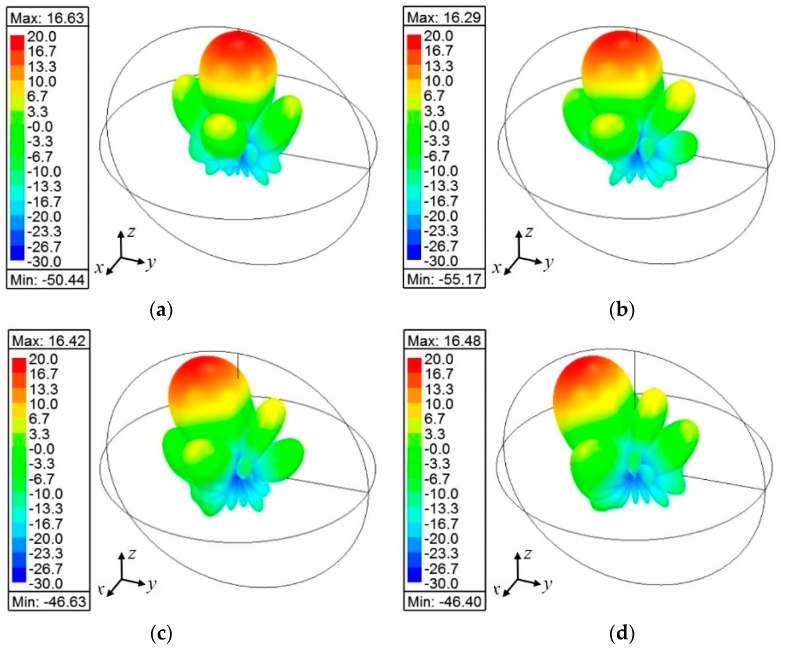
Simulated radiation patterns of the 4 × 4 patch antenna array when input phase offsets are (**a**) 0°, (**b**) 45°, (**c**) 90°, and (**d**) 135°.

**Figure 11 sensors-19-03982-f011:**
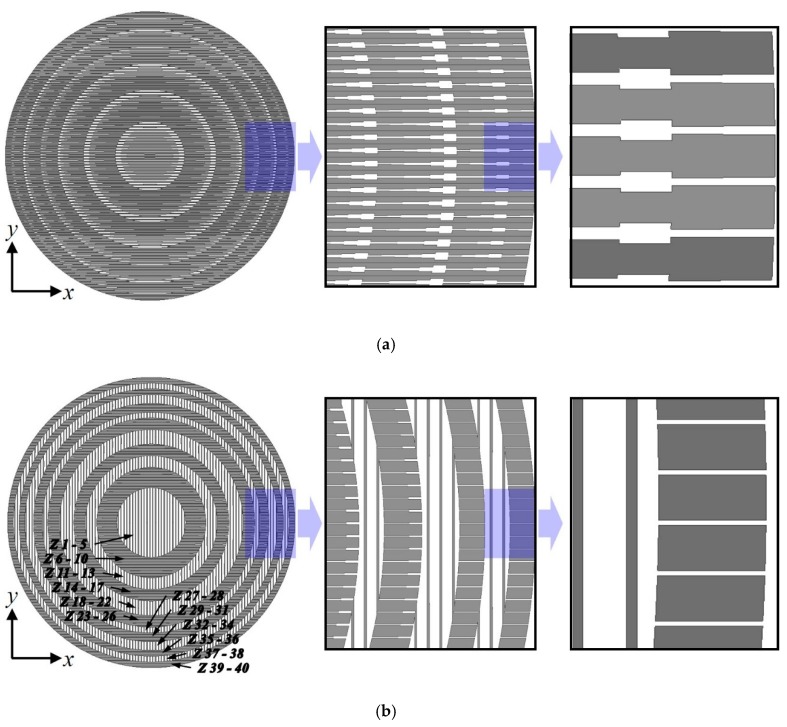
Top views and their zoomed views of (**a**) first and (**b**) second metal layers configured using coplanar capacitive and inductive strips.

**Figure 12 sensors-19-03982-f012:**
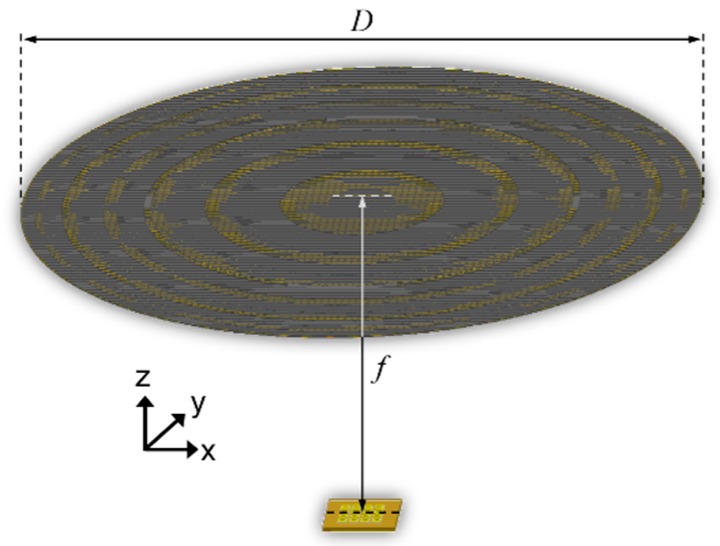
Oblique view of the designed lens.

**Figure 13 sensors-19-03982-f013:**
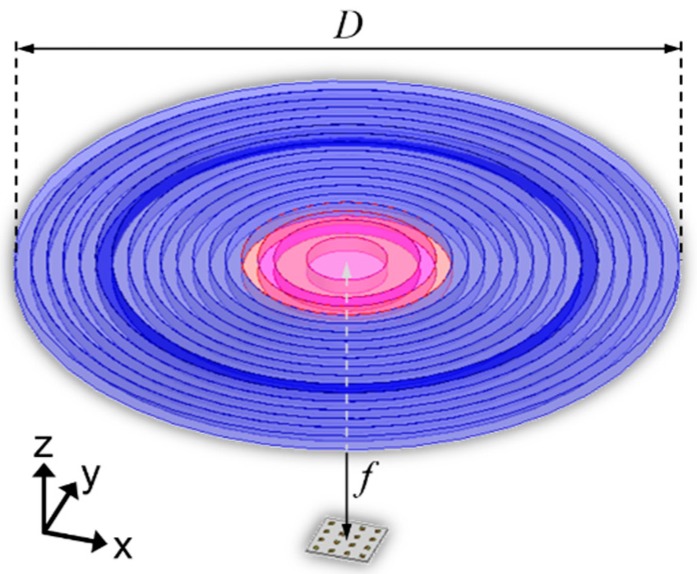
Oblique view of designed lens where the unit cells are replaced by effective mediums to expedite simulation time.

**Figure 14 sensors-19-03982-f014:**
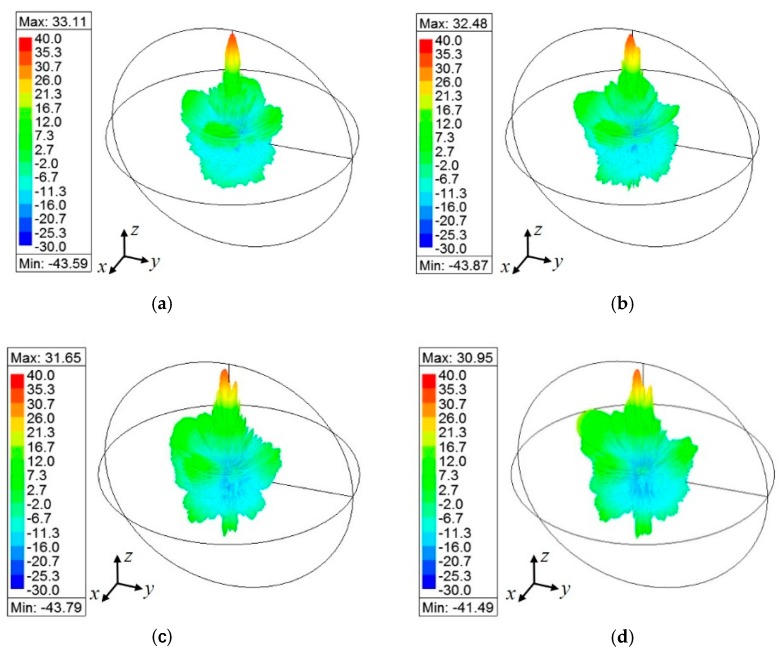
Simulated 3D radiation patterns for the input phase offset of (**a**) 0°, (**b**) 45°, (**c**) 90°, and (**d**) 135°, along *y*-axis for lens applied to 4 × 4 antenna array.

**Figure 15 sensors-19-03982-f015:**
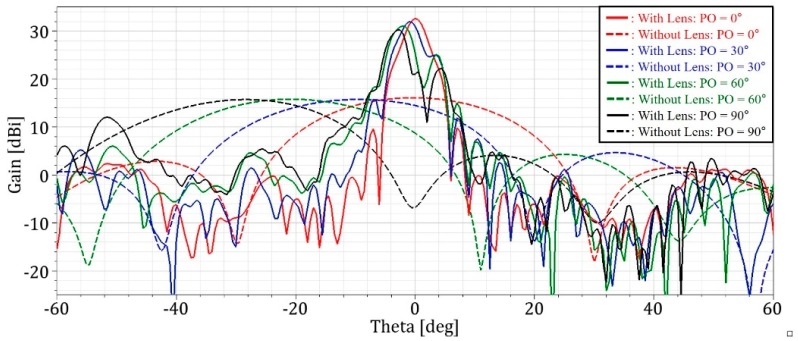
Simulated radiation patterns for 4 × 4 antenna array combined with the lens in case of each input phase offset.

**Figure 16 sensors-19-03982-f016:**
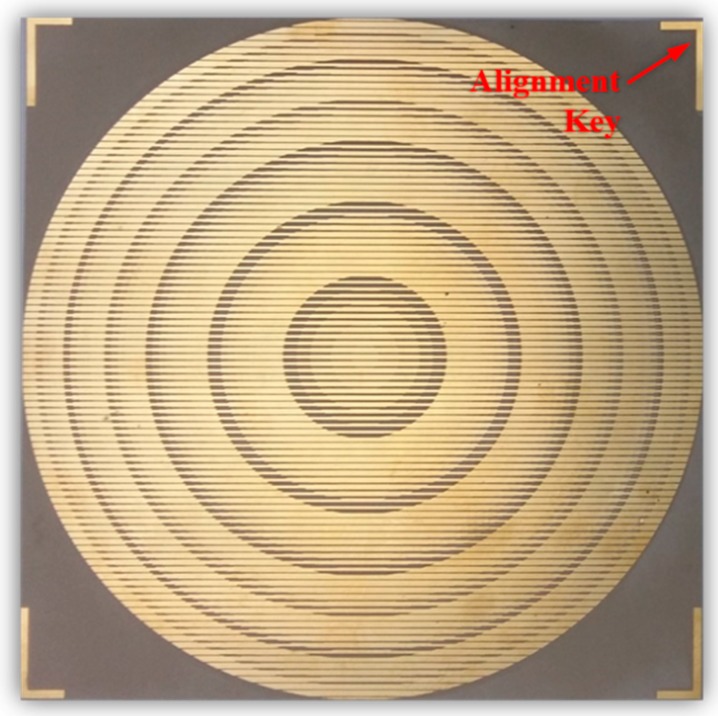
Top view of the fabricated sample of the proposed lens.

**Figure 17 sensors-19-03982-f017:**
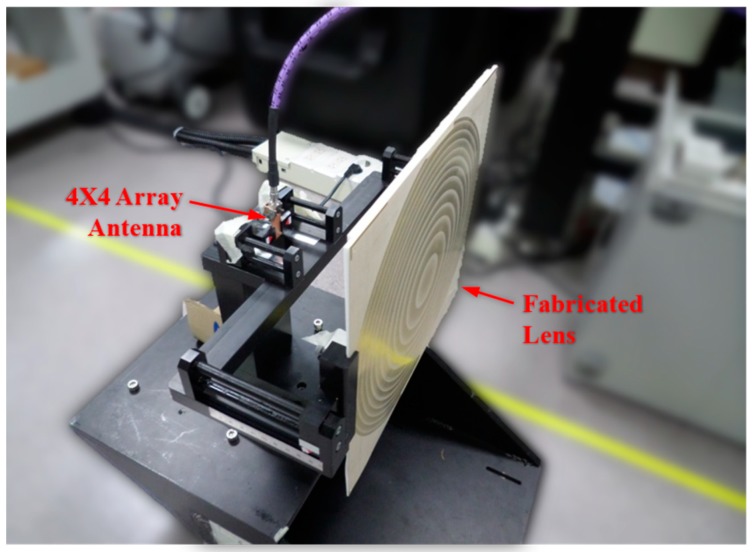
Measurement system and jig for the proposed lens.

**Figure 18 sensors-19-03982-f018:**
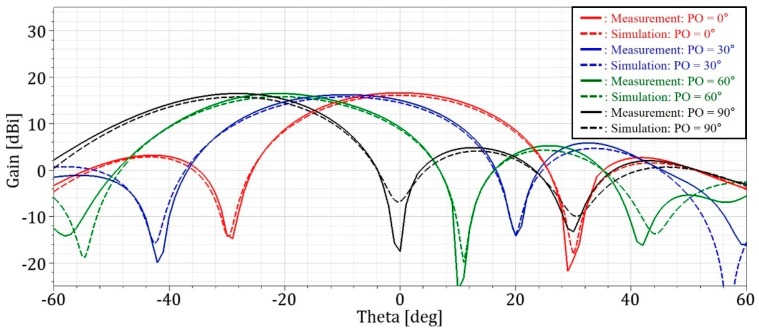
Simulated and measured radiation patterns of 4 × 4 antenna array in the absence of the lens.

**Figure 19 sensors-19-03982-f019:**
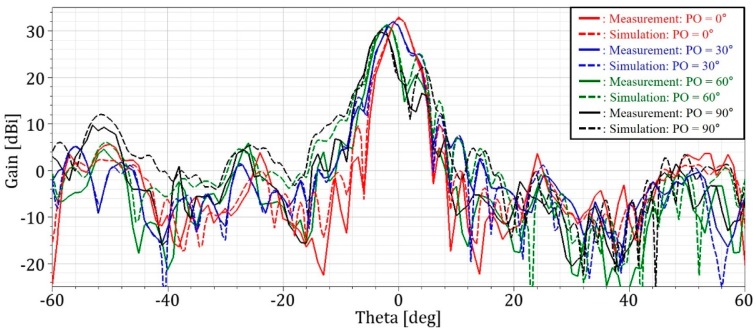
Simulated and measured radiation patterns of 4 × 4 antenna array in the presence of the lens.

**Table 1 sensors-19-03982-t001:** Geometric parameters, insertion loss, and phase shift of low-pass and band-pass unit cells at 60 GHz with normal incidence.

UC#	Filter Type	*g*1 (mm)	*g*2 (mm)	*g*3 (mm)	*w* (mm)	Insertion Loss (dB)	Phase Shift (°)
LP1	Low-pass	0.2	0.125	0.225		0.8	−276
LP2	Low-pass	0.225	0.15	0.25		0.5	−236
LP3	Low-pass	0.25	0.15	0.3		0.4	−208
LP4	Low-pass	0.3	0.25	0.3		0.2	−167
LP5	Low-pass	0.325	0.3	0.35		0.4	−142
BP1	Band-pass	0.325		0.375	0.25	0.2	−115
BP2	Band-pass	0.4		0.4	0.25	0.3	−98
BP3	Band-pass	0.5		0.6	0.3	0.4	−63
BP4	Band-pass	0.7		0.7	0.3	0.2	−27
BP5	Band-pass	0.9		0.95	0.25	0.7	−2

**Table 2 sensors-19-03982-t002:** Numbered zones, required phase shift at the center of each zone, selected UC# for each zone, incident angle, insertion loss and phase at the center of each zone.

Zone #	Number of Cascaded Unit Cells	Required Phase Shift in the Middle of Each Zone (deg.)	Selected UC #	Zone #	Number of Cascaded Unit Cells	Required Phase Shift in the Middle of Each Zone (deg.)	Selected UC #
1	2	−108	BP1	21	1	−16	BP4
2	3	−96	BP2	22	1	22	BP5
3	3	−64	BP3	23	1	66	LP1
4	2	−24	BP4	24	1	−249	LP2
5	2	−18	BP5	25	1	−188	LP3
6	2	−67	LP1	26	1	−130	LP5
7	2	−235	LP2	27	1	−74	BP3
8	1	−191	LP3	28	1	−9	BP4
9	1	−154	LP4	29	1	54	LP1
10	1	−121	LP5	30	1	−241	LP2
11	1	−83	BP2	31	1	−172	LP4
12	1	−45	BP3	32	1	−1058	BP1
13	2	−5	BP5	33	1	−38	BP3
14	1	78	LP1	34	1	31	BP5
15	1	−240	LP2	35	1	−256	LP1
16	1	−194	LP3	36	1	−183	LP3
17	1	−154	LP4	37	1	−111	BP1
18	1	−114	BP1	38	1	−41	BP3
19	1	−77	BP2	39	1	32	LP1
20	1	−45	BP3	40	1	−251	LP2

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
