# Peer review of "Affordable Thin Lens Using Single Polarized Disparate Filter Arrays for Beyond 5G toward 6G"

_sensors, 2019, doi:10.3390/s19183982_

Round 1

Reviewer 1 Report

Great article covering a very important and relevant topic on high-frequency ranges - construction feasibility. The proposed method proved viable and doable.

As a suggestion, the plots could be made more legible - the fonts are too small, legends in figs. 18 and 19 are not readable.

More details on the patch array could also be shown, for instance, the feed connectors name, the VNA used, if all ports were terminated in 50 ohms when measuring it etc.

I also feel there'd be another figure /detail of the lens zoomed in, in addition to those pics in fig. 11- a cut of a small piece of it, to see these layers and the geometry, it would make it more easy to understand.

Finally more words about HFSS simulation - was it run only with FEM method? Was used some hybridization?

Author Response

Response to Reviewer 1 Comments

Great article covering a very important and relevant topic on high-frequency ranges - construction feasibility. The proposed method proved viable and doable.

As a suggestion, the plots could be made more legible - the fonts are too small, legends in figs. 18 and 19 are not readable.

- Response: Thanks for the detailed observation. As per Reviewer #1's request, the figures (Fig. 15, Fig. 18 and Fig. 19 in the revised manuscript) are improved.

More details on the patch array could also be shown, for instance, the feed connectors name, the VNA used, if all ports were terminated in 50 ohms when measuring it etc.

- Response: Thanks for the useful comment. To reflect Reviewer #1’s request, authors revised Chapter 3 in the manuscript, as follows.

The antenna array designed in Section 2.3 is fabricated by adding a feed line that includes power dividers and phase shifting lines for each antenna element. The feedline is soldered by a Rogenberger 18S102-40ML5 connector for measurement.

Finally, the antenna array is connected to the Anritsu MS4647B Vector Network Analyzer to measure radiation patterns of the proposed lens antenna.

I also feel there'd be another figure /detail of the lens zoomed in, in addition to those pics in fig. 11- a cut of a small piece of it, to see these layers and the geometry, it would make it more easy to understand.

- Response: Thanks for the suggestion. Authors have agreed with your concern and improved Figure 11.

Finally more words about HFSS simulation - was it run only with FEM method? Was used some hybridization?

- Response: Thanks for the observation. In this study, the full-wave simulation was conducted only using FEM method of ANSYS HFSS.

Reviewer 2 Report

This paper proposes a design approach to obtain higher values of capacitance and inductance for a high-gain thin lens. Some detailed comments for the paper are listed as follows:

(1) Explain in detail why the higher values of capacitance and inductance are needed in Section 2.1.

(2) Provide the values of Lx, Ly, Sx, Sy, Ux, Uy, Px, Py in Figure 9 and the operating frequency.

(3) Adjust the maximum values of the color scale bars in Figure 10.

(4) Show the zones(#1~#40) of Table 2 in Figure 11.
